# Agricultural Support and Public Policies Improving Sustainability in Brazil's Beef Industry

Luana Molossi [1,*] , Aaron Kinyu Hoshide [1,2] , Daniel Carneiro de Abreu [1,3] and Ronaldo Alves de Oliveira [1]

1   AgriSciences, Universidade Federal de Mato Grosso, Caixa Postal 729, Sinop 78550-970, MT, Brazil
2   College of Natural Sciences, Forestry and Agriculture, The University of Maine, Orono, ME 04469, USA
3   Instituto de Ciências Agrárias e Ambientais (ICAA), Universidade Federal do Mato Grosso,
     Campus Universitário de Sinop, Avenida Alexandre Ferronato, 1200, Sinop 78550-728, MT, Brazil
*   Correspondence: luana_molossi@hotmail.com; Tel.: +55-66-99698-1573

**Abstract:** Since the dawn of Brazilian trade, extensive cattle farming has predominated. Brazil's extensive pasture-based system uses pasture plants adapted to climate and soil conditions with limited use of purchased inputs. However, new technologies such as integrated crop and livestock systems have recently been adopted, with government support and public policies that are intended to encourage increased agricultural production in Brazil. Domestic and international stakeholders have prioritized sustainable agricultural development in Brazil's beef sector to reduce deforestation and other natural-habitat conversions. This review provides an overview of beef production in Brazil, focusing particularly on (1) historical factors that have encouraged an extensive, low-intensity style of production and (2) how national public policies supporting agriculture have improved sustainability in Brazil's beef industry. Since the beginning of the twenty-first century, specific public policies for rural areas began to implement changes that addressed environmental concerns. Programs aimed at protecting secondary forests and increasing their areas are needed to offset the 42% of Brazil's greenhouse gas emissions that come from land-use change. To produce more beef with less environmental impact, cattle ranchers need to use their land more productively. Thus, public policy initiatives need to combat deforestation and preserve the environment and local communities, while sustainably intensifying Brazil's beef production.

**Keywords:** Amazon; beef; Brazil; deforestation; environmental impacts; greenhouse gases; livestock intensification strategies

## 1. Introduction: History of Cattle Breeding and Production Systems in Brazil

The growth of beef cattle in Brazil has solidified the country in international markets as one of the largest exporters of beef. In 2021, the Brazilian herd was estimated at 196.47 million head, with 39.14 million head slaughtered. The volume of meat produced was 9.71 million metric tons of carcass-equivalent weight. Of this total volume, 25.51% of Brazil's beef production—2.48 million metric tons—was exported, while 7.24 million metric tons—equivalent to 74.49% of Brazil's beef production—were destined for the domestic market [1]. Brazil's beef production has historically and currently been dominated by an extensive pasture-based system, in which animals typically take two to four years to reach slaughter weight [2].

Brazilian cattle are predominantly tropic breeds (*Bos indicus*, such as the Nelore breed), with temperate breeds (*Bos taurus*) more prevalent in southern Brazil. During their evolution, *Bos indicus* cattle acquired genes that confer a greater thermotolerance in response to heat stress than that of European breeds. This is one of several reasons that *Bos indicus* (e.g., Nelore) are the predominant cattle in central Brazil, which has high temperatures and a dry climate throughout the year [3]. *Bos taurus* cattle are generally more adapted to environments with milder and more humid temperatures, such as those as found in the

southern region of Brazil [4]. Cattle gain weight during the wet season (October through March) but lose weight during the dry season (April through September) as pasture productivity diminishes (Figure 1). Brazil's pastures comprise approximately 151 million hectares, including areas that are both natural and cultivated (Figure 2).

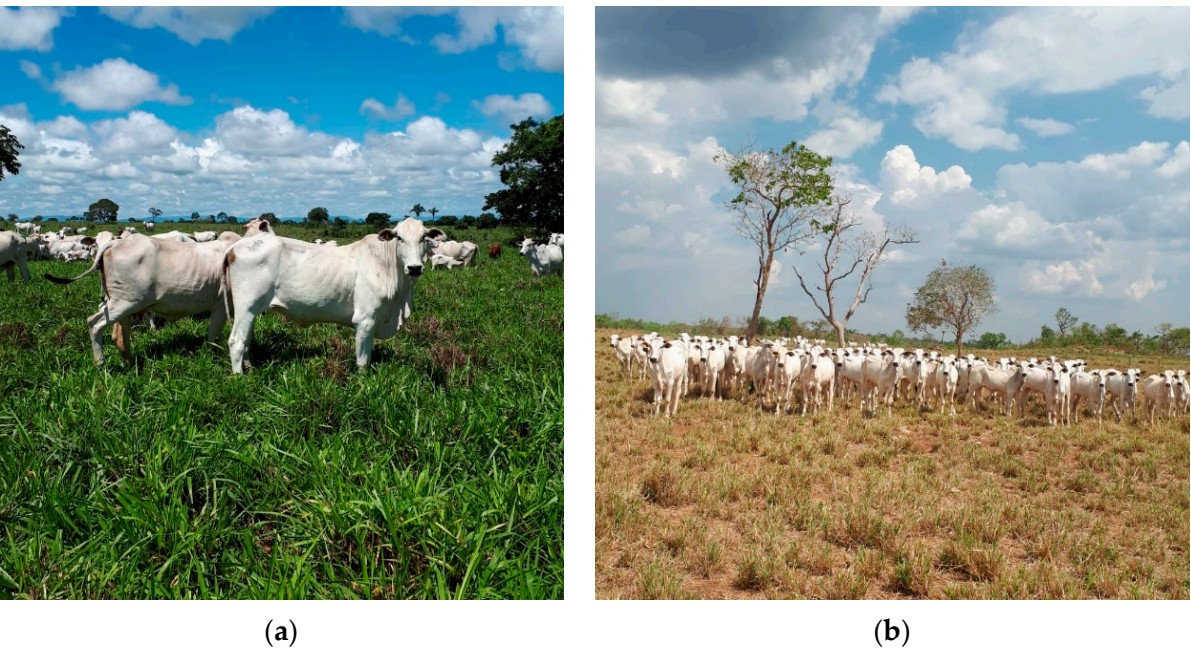

(**a**)   (**b**)

**Figure 1.** Beef cattle (*Bos indicus*, such as the Nelore breed) grazing on extensive pasture (*Brachiaria* spp.) during the (**a**) wet season and (**b**) dry season in midwest region of Brazil (Source: corresponding author).

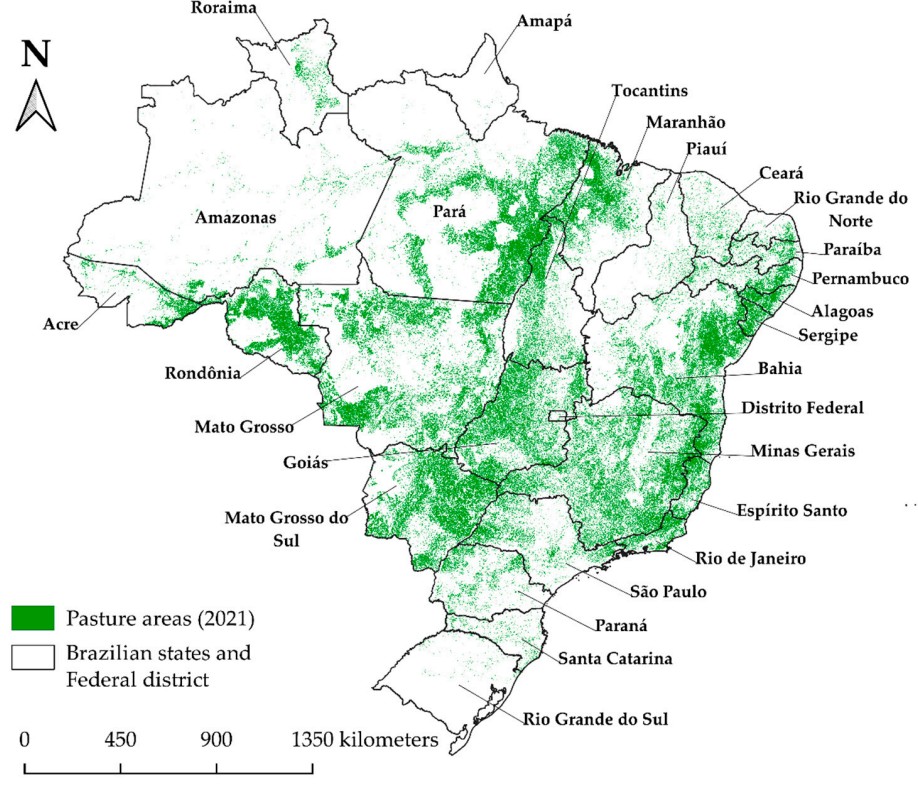

**Figure 2.** Spatial distribution of pasture areas in Brazil in states and the federal district in 2021 [5].

However, Brazil's current beef production, trade, and marketing are completely different from those practices in Brazil's beef industry 40 years ago. Then, the total beef herd was less than half of the current total, and beef production did not completely meet the Brazilian population's consumer demand [6]. In 2021, the beef production of 9.71 million metric tons of carcass-equivalent weight [1] was enough to meet the domestic demand for beef, which was 36.4 kg per person per year [7]. Even with more recent increases in beef production, the total pasture area associated with beef cattle has declined due to intensification strategies used in Brazil's beef production systems (Figure 3).

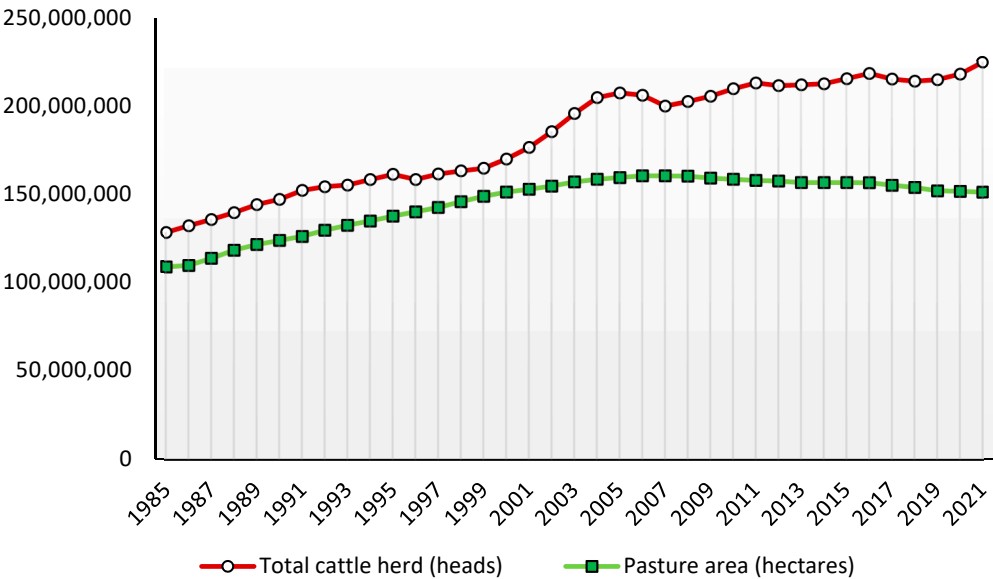

**Figure 3.** Total Brazilian cattle herds and pasture areas from 1985 to 2021 [8].

The first cattle arrived in Brazil in 1533, during the establishment of the first Portuguese colony on the island of São Vicente in the state of São Paulo [9]. In the middle of the 16th century, the Portuguese royal court encouraged the export of cattle to the Bahian Recôncavo region in northeastern Brazil. Gradually, with the growth of the economy in coastal areas, cattle raising expanded into the country's interior [10]. Since these commercial beginnings, Brazil's beef production has relied on an extensive pasture-based system, using plants adapted to local climate and soil conditions, with limited use of inputs [11].

With the opening of the Brazilian economy and the greater financial support of the agricultural sector in the 1990s, profound changes in Brazil's beef industry took place thereafter [12]. The development of practices aimed at increasing productivity has led to increases in intensive production systems in some regions. These technologies involve the genetic improvement of the animals, control of the economic management of the property, and a supply of concentrated feed for the animals, using feedlots or semi-feedlots to reduce the time to slaughter and increase profitability [13]. Thus, there has been a recent increase in cattle herd size (Figure 4), together with increases in cattle stocking density (head/hectare) (Figure 5), in particular regions in Brazil. Regions with such increases include the northern and central parts of Brazil, such as Brazil's center-west and north regions. While cattle herd numbers have stayed relatively stable from 1985 to 2021 in Brazil's northeastern, southeastern, and southern regions, the numbers have expanded in the north (from 5.3 to 55.7 million) and center-west (from 41.1 to 75.4 million) over these 36 years (Figure 4). Stocking densities of cattle are relatively high (>1 head/hectare), except for states along Brazil's southeastern coast. The adoption of the new technologies that allow for increased productivity and profitability in the beef cattle industry was possible due to the support and public policies developed for agricultural production in Brazil, with a focus on these developments occurring in a sustainable way [14].

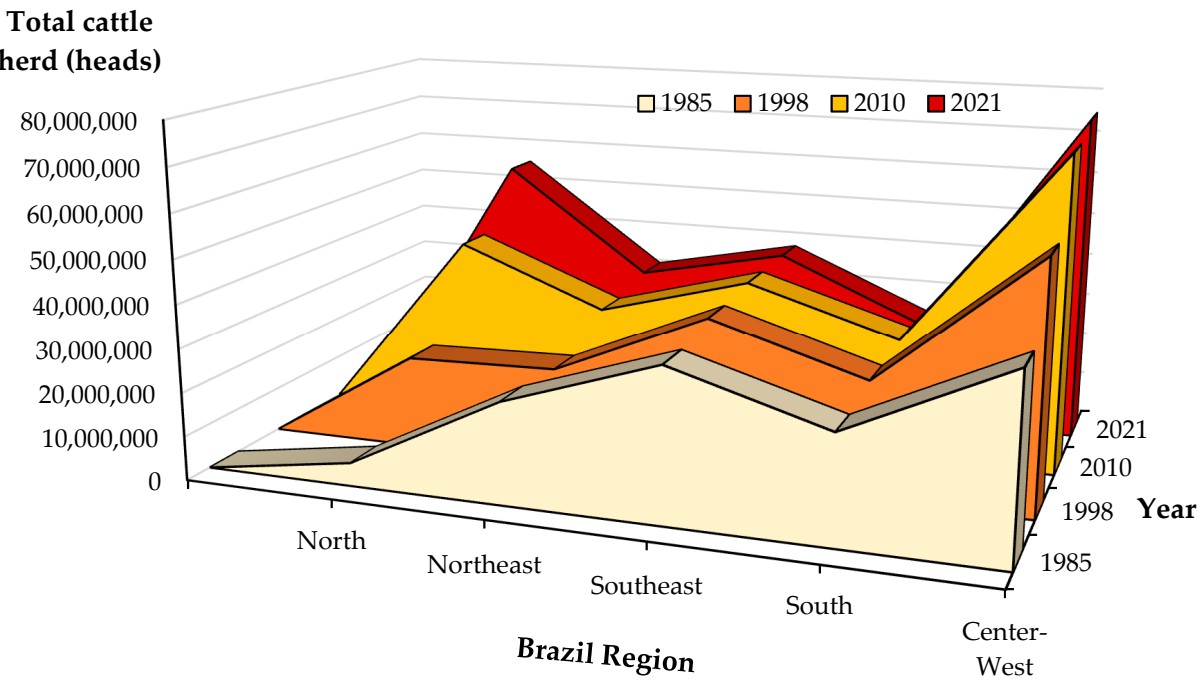

**Figure 4.** Cattle herd sizes (heads) in the northern, northeastern, southeastern, southern, and center-west regions of Brazil in the years 1985, 1998, 2010, and 2021 [8].

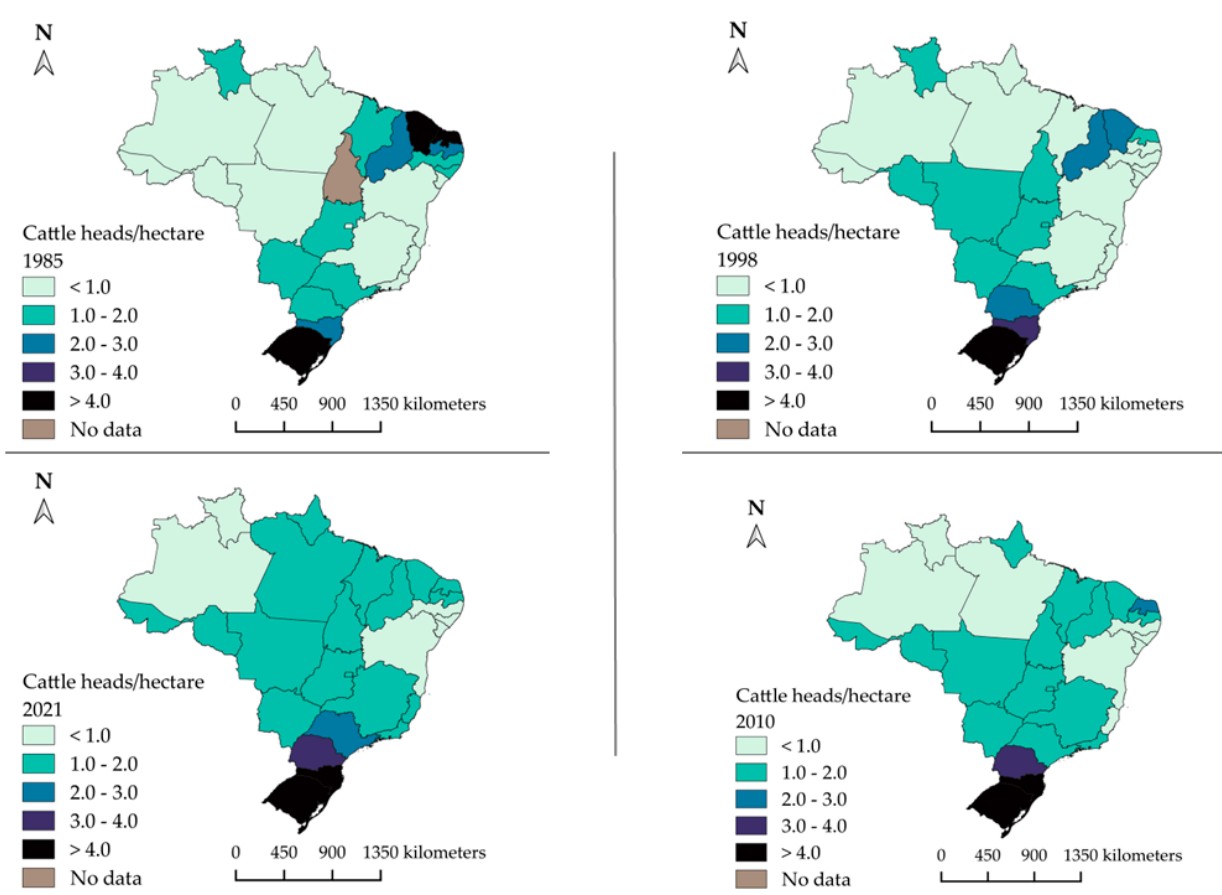

**Figure 5.** Cattle stocking densities (heads/hectares) by Brazilian states in 1985, 1998, 2010, and 2021 [15].

Such incentives have occurred because extensive systems have low efficiency, as they are based on cattle production with low technological intensity and management standards [13]. Thus, extensive systems for producing Brazilian beef have low productivity, requiring large amounts of land for grazing on seeded pastures that, over time, can become over-grazed and degraded [16]. Amazon rainforest deforestation, which precedes the establishment of pastures, has focused international attention on reducing deforestation [7]. This attention has encouraged three types of intensification in Brazil's beef-production systems to produce more beef on already existing pastureland: (1) re-seeding degraded pastures, (2) feeding grains in pastures, and (3) semi-intensive feedlots [17]. Pasture re-seeding involves the new establishment of pastures through forage sowing and fertilization to meet the plant requirements that are needed to optimize production [17]. Beef cattle can be fed grain at feeding stations in pastures that typically have low nutritional value [18]. Feedlots typically supply concentrated feeds for finishing animals, where the feeds normally consist of an energy and protein source [19]. Management-intensive rotational grazing—where cattle are rotated daily between paddocks that have been created by portable, electrified poly-wire—is much less popular [20], reinforcing producers' preferences for management systems that take less time.

Thus, the goal of this review is to provide an overview of beef production in Brazil, focusing particularly on (1) historical factors that have encouraged an extensive, low-intensity style of production and (2) how national agricultural support public policies have improved sustainability in Brazil's beef industry in order to better balance agricultural economic growth while reducing the environmental impacts of beef cattle. Such public policies have directly targeted the beef sector or indirectly targeted beef sustainability by focusing on reducing deforestation. We also highlight prospects for Brazil's beef production chain and compliance with market requirements. Current and future strategies for the sustainable intensification of Brazilian beef production systems are also explored. These strategies can reduce Brazilian beef's long shadow over the Amazon rainforest by sustainably intensifying land that has already been cleared, without further deforestation [21,22].

## 2. Historical Factors Encouraging Extensive-Pastured Beef

The agricultural sector has been important for the Brazilian economy since the beginning of colonization [21]. However, Brazilian beef cattle producers are historically characterized by resistance to technological innovations and by more primitive management, which negatively characterized the activity over the past several decades [22]. At the beginning of Brazil's colonization, the coastal lands were used to produce sugarcane, which was the main economic activity. Therefore, livestock was relegated to the interior of the country. In Brazil's northeastern region, livestock was concentrated in the back country (*sertão*) that supplied the northeastern coast from Maranhão state to Bahia state. In the south of the country, livestock farming in Brazil's pampas grassland biome is distinct from that in the rest of the country, since beef production in this region is based on the use of highly diversified native pastures (in contrast to exotic introduced pastures). If animals are not over-stocked. then livestock can contribute to pasture conservation via more sustainable management of these grassland biome agro-ecosystems [23].

However, despite developing more autonomously in the southern region than in the northeast, the growth in livestock industries was not continuous nor consistent [10]. The stagnation of livestock growth was also attributed to the fact that cattle served as a capital reserve during Brazil's inflationary periods. In addition, due to the extensive-exploration model and the availability of large areas of land for exploration, beef cattle were historically used as a land claim to open new areas on the agricultural frontier, while already-established areas were converted to agriculture uses [24]. This extensive-grazing strategy was characterized by low livestock productivity. Generally, meat producers occupy frontiers and use the land for extracting nutrients from the soil without replacing them. Instead of maintaining pasture quality and re-seeding degraded pastures, new areas are deforested, either on farms themselves or in other areas, such as new frontiers where

beef production is being established. Such factors are predominant in Brazil's extensive pasture-based system, leading to pasture degradation as well as soil degradation and compaction [13,25].

Additionally, the state and federal governments have implemented programs to regulate illegal possessions of land through donations or sales of lands at below-market prices, encouraging speculative land occupations. These land occupations favor the formation of small unproductive properties [25], which results in negative consequences for the sustainability of livestock, such as low zootechnical indices, environmental impacts, and reduced economic returns [26]. Livestock activities continue to be practiced, to a large extent, within the traditional, extensive system [27]. However, with increases in the demand for food and the technological advancements in agriculture, new production techniques have been introduced for raising cattle, such as integrated systems [28].

Integrated crop and livestock systems can be used in degraded pastures, reducing the need for agricultural expansion. In addition to promoting the increase and diversification of production, integrated systems can enhance carbon stock and soil fertility [29]. However, integrated crop and livestock systems are limited in Brazil [30]. Like crop-livestock integration, rural development is a multidimensional process involving a wide range of actors, institutions, and institutional infrastructure. This can range from communities and farmers to public-policy makers, passing through organizations representing broad social and productive sectors, as well as science and technology institutions across different levels [31]. Public policies are actions and decisions formulated in different spheres of legislative and executive power for the purpose of solving public problems. The process of formulating a public policy begins with the detection of an element with respect to which the government must act. The process must be configured as identifying a problem to be solved, understanding why it is important to provide solutions to such a problem, and anticipating the expected results of adopted solutions [32].

Changes in public policies prioritizing environmental concerns began to be implemented in Brazil, especially for family farming, starting in the early 2000s [33]. Programs that aim to protect secondary forests and increase their areas are necessary to offset Brazil's greenhouse gas emissions, approximately 42% of which come from global land-use change [34]. It is expected that increased beef cattle production will occur via increases in productivity rather than by expansion of pasture areas, transforming current extensive systems into systems with greater livestock intensification [35]. Such strategies will also result in a significant net reduction in greenhouse gas emissions [36]. Thus, it is essential to implement public policy mechanisms that counter environmental degradation and encourage the conservation of natural biomes and the sustainable use of natural resources, while allowing for gains in productivity [37]. This approach can counter illegal land occupation practices, as well as speculative and destructive actions in forest preservation areas [38].

## 3. Recent Public Policies for More Sustainable Livestock in Brazil

In this section, we discuss two recent types of sustainable agricultural development policies in Brazil. In Section 3.1, we cover agricultural policies that have had a direct impact on improving sustainable intensification (SI) in Brazil's beef-production industry. In Section 3.2, we highlight environmental public policies that have had an indirect impact on the SI of Brazil's beef cattle herd. These environmental policies have typically preceded the more direct policies discussed in Section 3.1 and have involved a reduction of Amazon deforestation and a conversion of land use in other natural Brazilian habitats for cattle pastures.

### 3.1. Agricultural Policies Directly Supporting Sustainable Livestock

We ordered recently enacted public policies in Brazil from the least challenging to the most challenging for producers to adopt. Most of these policies, with the exception of the 1965 Brazilian Forest Code, have been adopted over the past two decades. It is important to note how involved beef producers have been with these programs and to what extent

these policies encourage sustainable intensification strategies for Brazil beef production. These strategies include good agricultural practices, low-carbon production, integrated crop–livestock–forest systems, pasture-based grain supplementation, pasture rehabilitation, and semi-intensive feedlots.

### 3.1.1. Agriculture and Livestock Plan (Plano Agrícola e Pecuário)

Brazil's Agriculture and Livestock Plan is the main instrument for directing public policies aimed at the agricultural sector. The plan includes measures to encourage the production of certain products, while providing resources for agricultural producers, including credit at favorable interest rates that is made available throughout the harvest year. Brazil's harvest year runs from July to June. The amount of money devoted to the Agriculture and Livestock Plan depends on the budget of the National Treasury and the amount allocated to financial subsidies for the agricultural sector [39].

The Agriculture and Livestock Plan embodies the main measures to support commercialization, rural risk management, and credit support. Thus, government actions are necessary to ensure the continuity of the advances that have been already achieved in increasing agricultural productivity. The Agriculture and Livestock Plan also can sustain the income of rural producers and ensure the flow of food, fuel, and fiber to both domestic and international markets. Brazil has had favorable conditions related to production costs, which have increased the competitiveness of its agricultural exports [40].

The annual publication of the previous Crop Plan became a tradition, dealing only with questions related to crops and marginalizing the livestock sector. In 2000, thanks to requests from entities representing milk producers and their cooperatives and to the sensitivity of the Brazilian government, the famous Crop Plan proposed measures related to dairy production for the first time. In the same year, the previous Crop Plan was renamed the Agriculture and Livestock Plan to definitively address livestock via announced measures [41].

The Brazilian Agricultural Research Corporation (Embrapa), which is considered to be an important developer of technologies in the Brazilian agricultural sector, is mentioned in the Agriculture and Livestock Plan 2012–2013. Among the highlights of Embrapa's programs is the Good Agricultural Practices Program (GAPP) for beef cattle. The GAPP is not an agricultural credit measure, but rather a mechanism within the plans that can differentiate access to credit by rural producers. Created in 2005, this program encompasses a set of norms and procedures that must be observed by rural producers in order to make their properties more sustainable. Various factors, such as the management and the social function of rural properties, human resources management, environmental management, rural facilities, pre-slaughter management, animal welfare, pastures, food supplementation, animal identification, sanitary control, and reproductive management, are crucial for the effectiveness of GAPP's adoption on farms [42].

### 3.1.2. ABC Plan, or Low-Carbon Agriculture Plan (Agricultura de Baixa Emissão de Carbono)

The Sectorial Plan for Mitigation and Adaptation to Climate Change for the Consolidation of a Low-Carbon Economy in Agriculture, also known as the ABC plan, is one of the sectoral plans prepared in accordance with Article 3 of Decree No. 7.390/2010. Its purpose is to organize and plan actions to be carried out for the adoption of sustainable production technologies. These sustainable technologies are selected with the objective of responding to Brazil's commitments to reduce greenhouse gas (GHG) emissions in the agricultural sector [43]. The ABC plan addresses climate change, ecosystem and biodiversity management, resource use efficiency, and sustainable consumption/production, in addition to presenting guidelines for environmental governance, thereby contributing to the exchange of information and experiences among the public, private, and academic sectors [44].

The ABC plan is composed of seven programs, six of which refer to mitigation technologies, and another that includes actions needed to adapt to climate change. The first program involves rehabilitating degraded pastures. The second program is the integration

of crop–livestock–forest (ICLF) systems and agroforestry systems (AFSs), which involves integrating commercial forestry species (e.g., *Eucalyptus* sp.) with commodity crops, such as soybeans (*Glycine max* L.), corn (*Zea mays* L.), and cotton (*Gossypium* sp.), and livestock, such as beef (*Bos indicus*, such as Nelore cattle). The third and fourth programs comprise the direct planting system (DPS) and biological nitrogen fixation (BNF) programs, which involve soil mobilization only in a sowing line or planting hole, the permanent maintenance of soil cover, species diversification, and increased fertilization efficiency. The fifth program involves re-forestation, while the sixth program focuses on the treatment of animal waste. Finally, the seventh program addresses climate-change adaptation [45].

The ABC program provides agricultural producers with opportunities to incorporate sustainable technologies into their production processes for more efficient production. This can increase income through increased productivity and product diversification. It also can mitigate environmental liabilities, reduce pressure on native forests, and lower GHG emissions, thereby enhancing sustainable agricultural production of food for local Brazilian and export markets. This new sustainable agricultural program involves government incentives that provide attractive alternatives to existing financing instruments in the marketplace [46]. With the adoption of more sustainable techniques and production systems, it is possible to increase productivity, reduce deforestation, reconcile soil and water conservation, adapt rural properties to environmental legislation, expand the area of cultivated forests, and encourage the recovery of degraded areas [47].

### 3.1.3. National Integrated Crop–Livestock–Forest Policy (Política Nacional de ICLF)

The silvopastoral system is a technological option for integrated crop–livestock–forest integration that consists of an intentional combination of trees, pastures, and cattle in the same area at the same time. The approval of Law 708/07 on 4 February 2013 established the National Integrated Crop–Livestock–Forest Integration (ICLF) policy in Brazil. The ICLF policy reinforces the growing interest in the use of sustainable production systems. This law integrates agricultural and forestry activities carried out in the same area, in consortium, in succession, or in rotation. It seeks synergistic effects between the components of the agroecosystem, with objectives of recovering degraded areas, and enhancing economic viability, and supporting environmental sustainability [48].

The ICLF policy is a production strategy that includes the economic, social, and environmental aspects of sustainability. With the growing concern about the relationship between the environment and livestock, the challenge of establishing sustainable production systems is paramount. Silvopastoral systems are capable of meeting this challenge [49].

The ICLF policy's core principles involve the preservation and improvement of the soil's physical, chemical, and biological conditions and compliance with environmental protection laws. Cooperation between the public and private sectors and non-governmental organizations is recommended to foster the diversification of economic activities. Another policy guideline is the encouragement of direct planting in crop residue from the preceding crop as a soil-conservation management practice [50]. The ICLF policy also aims to mitigate deforestation caused by land-use conversion of native vegetation into pastures and/or crops and contributes to the maintenance of permanent preservation areas and legal reserves. The recovery of degraded pasture areas is also encouraged via sustainable production systems, such as the adoption of conservation practices and agricultural systems that maintain higher levels of organic matter in the soil and that reduce greenhouse gas emissions [51].

### 3.1.4. Agriculture Modernization and Natural Resources Conservation Program (Programa de Modernização da Agricultura e Conservação de Recursos Naturais—Moderagro)

The Moderagro (the Program for the Modernization of Agriculture and Conservation of Natural Resources) aims to support and encourage the production, processing, industrialization, packaging, and storage of agricultural products. This program is a Banco Nacional do Desenvolvimento (BNDES) project that enables rural producers to finance actions to recover soils, defend animals, acquire and apply agricultural fertilizers, and

build facilities for the storage of agricultural machinery and implements, as well as for the storage of inputs [52]. The program supports and encourages the sectors of production, processing, industrialization, packaging, and storage of animal products from the beekeeping, aquaculture, poultry, chinchilla, rabbit, sheep and goat, frog, pig, and dairy farming industries. The agricultural production of floriculture, fruits, olives, horticulture, palm trees, yerba mate, nuts, and fishing are also encouraged [53].

### 3.2. Agricultural Policies Indirectly Supporting Sustainable Livestock

Other recently enacted environmental policies in Brazil have had more of an indirect impact of improving the sustainability of Brazil's livestock. These public policies have reduced Amazon deforestation and Cerrado habitat conversion. In general, these public policies were enacted earlier than policies that directly focus on livestock (Figure 6). We discuss whether beef producers were engaged and involved in the writing and implementation of these public policies. We also highlight how influential the limitation of grazing areas for cattle by preserving native habitat has been in encouraging beef producers to sustainably intensify their production systems.

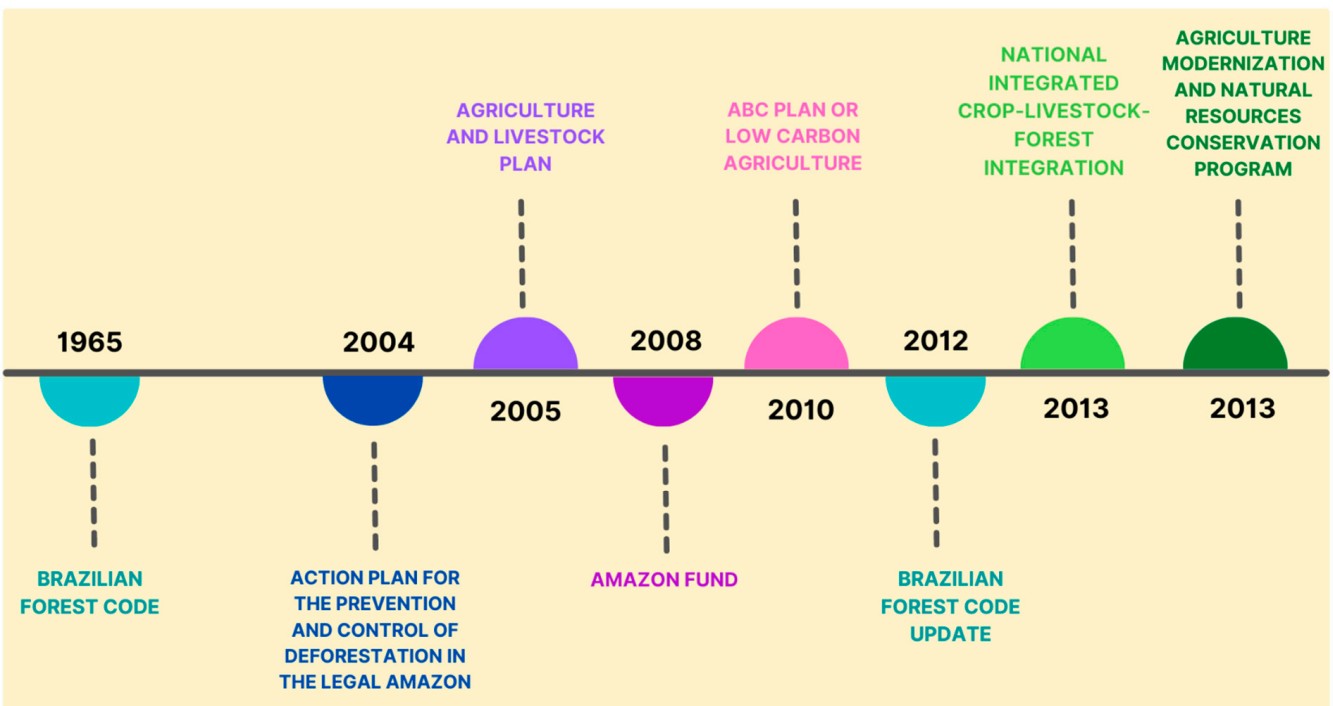

**Figure 6.** Historical timeline of direct and indirect public policies impacting sustainability in Brazil's beef cattle industry from 1965 to 2013.

### 3.2.1. The 1965 Brazilian Forest Code & 2012 Update (1965 & 2012 Código Florestal Brasileiro)

The Brazilian Forest Code is an indispensable political instrument for managing the country's economic development. Focusing on the different historical periods in Brazil, the evolution of the Brazilian Forest Code (BFC) reflects past political, economic, and environmental events, as well as the development intentions articulated through the law [54]. The BFC was created in 1965 with the aim of preserving forests and streamlining their management. At the time the code was created, the main agricultural activities were coffee (*Coffea arabica* and *Coffea robusta*) and sugar cane (*Saccharum* spp.). The code also contained several provisions, such as the prohibition of occupying steep slopes and a determination for rural landowners to maintain a reserve of native vegetation on their farms to contribute to preserving existing forests [55].

Although instituted in 1965, it was only in the 1980s that the Legal Reserve and Permanent Preservation Areas were effectively introduced into law, via a provisional measure. Another important aspect of the 1965 code was the creation of the National System of Conservation Units, which covered the different types and categories of protected areas within a single management system [56]. Law 12,651, which dates from 25 May 2012, introduced a series of new forest regulations. The 2012 Forest Code, currently on the books, consolidated protected areas from the previous code and included conceptualizations/specifications for delimitation of each area provided for in the law [54].

The BFC laws' innovations resulted in the creation of the Rural Environmental Registry (RER), as well as implementation of the Environmental Regularization Program. Under the RER, it is possible for the federal government and state environmental agencies to determine the location of each rural property and the status of its adherence to environmental standards for the preservation of native vegetation on the property. Additionally, the new law authorizes a series of benefits for family farmers or owners of smaller properties by including their properties in the RER [57].

The RER is a nationwide electronic registration system for gathering data on rural properties/possessions that are used for environmental and economic planning and for combating deforestation. The RER is the first step enabling rural producers to show that their rural property complies with the Forestry Code. If an owner does not register in the system, the owner is prevented from having access to agricultural credit from financial institutions. In addition, it can be difficult for rural producers to sell their agricultural products, as some companies require RER documentation from producers in order to buy their products [58].

### 3.2.2. Plan to Prevent and Control Deforestation in the Brazilian Amazon

The Action Plan for the Prevention and Control of Deforestation in the Legal Amazon (APCDAm) was created in 2004 with the aim of continuously reducing deforestation and creating conditions for the transition to a sustainable development model in the Legal Amazon. One of the main initial challenges was to integrate the fight against deforestation into Brazilian State policies [40]. Thus, the APCDAm became a strategic initiative of the Brazilian government that was included in the guidelines and priorities of this sustainable development plan for the Amazon. Therefore, the problem of the Amazon became part of the political agendas at the highest levels of the federal government and ministries [32].

Because the fight against the causes of deforestation could no longer be conducted in isolation by environmental agencies, the complexity of the challenge required coordinated efforts from different sectors of the federal government [40]. The APCDAm is implemented by more than a dozen government ministries; it was coordinated by the Civil House until March 2013, and thereafter by the Ministry of the Environment. The APCDAm is structured to address the causes of deforestation in a comprehensive, integrated, and intensive way, with actions articulated around three themes: land and territorial organization, environmental monitoring and control, and the promotion of sustainable production [59].

### 3.2.3. Amazon Fund (Fundo Amazon)

The Amazon Fund aims to encourage Brazil and other developing countries that have tropical forests to maintain and increase voluntary reductions in the emission of greenhouse gases caused by deforestation and land degradation [60]. The Amazon Fund was created by Decree No. 6527 on 1 August 2008. This fund raises donations for non-reimbursable investments to prevent, monitor, and combat deforestation and to promote the conservation and sustainable use of forests in the Amazon biome. Its creation was a consequence of the success achieved by the APCDAm in reducing deforestation in the Amazon since its implementation in 2004. The creation and raising of resources by the Amazon Fund have led to funding Brazilian efforts to reduce the loss of forests via projects that work on this theme, in synergy with government agencies [59].

## 4. Comparison of Agricultural Public Policies

In Table 1, we summarize and compare the program impacts and costs of both the direct and indirect public policies that affect Brazil's beef industry. Direct public policies share three common themes. The first is to provide financial credit to agricultural producers in order to intensify their production in a sustainable way. The second is to increase production without the need to expand into new areas (e.g., without Amazon deforestation). The third shared theme is to reduce the environmental impacts of agricultural production. Indirect public policies support direct policies with the main objective of preserving the environment. These indirect policies can help reduce the adverse environmental impacts of meat production in Brazil. For example, the Amazon Fund finances programs that reduce Amazon rainforest deforestation and greenhouse gas emissions. The Amazon Fund's cost is high at USD 337.3 million in 2021 (see Table 1).

The costs, in 2021, of each program to finance farmers and ranchers varied. The Agriculture and Livestock Plan is the main public policy for providing rural credit in Brazil. Therefore, it is the policy that bears the greatest cost for the Brazilian government, at USD 48.3 billion in 2021 (see Table 1). On the other hand, the National Integrated Crop-Livestock-Forest Integration public policy had the lowest cost in 2021 (USD 13.1 million) as this policy specifically focuses on encouraging integrated crop–livestock–forestry production. This policy is unlike other policies, such as the Agriculture Modernization and Natural Resources Conservation Program, which can cover different forms of modernization within rural properties, thus requiring greater financial resources (USD 68.1 million) annually (see Table 1).

Sustainability challenges in the livestock industry require simultaneous progress in production and environmental performance [61]. The public rural credit policy in Brazil incentivizes rural producers to recover fragile areas and pastures, to reduce production in unproductive soils and degraded areas, to plant forests, and to preserve natural resources. This credit policy also encourages the implementation and improvement of agricultural production systems, such as organic livestock systems and direct-planting systems [62].

The Agriculture and Livestock Plan consolidates the main actions and public policies aimed at the agricultural sector, with an emphasis on rural credit [59]. Some credit programs, such as the Agriculture Modernization and Natural Resources Conservation Program, are aimed at innovation, such as the implementation of an animal traceability system for human consumption. However, credit programs can also support other on-farm investments, such as recovering soils by financing the acquisition, transport, application, and incorporation of agricultural fertilizers [63]. The ABC program is another important program for modernizing sustainable production systems and mitigating emissions through low-carbon agriculture [62]. The same is true for the National Integrated Crop–Livestock–Forest Integration program, which optimizes land use, raises productivity levels, diversifies production, and generates quality products via integrated systems [41].

Other policies provide security and support for direct credit to rural producers, such as policies that establish environmental criteria and inspection actions to encourage environmental improvements. Such policies must be evaluated against specific outcomes, such as targets related to deforestation and greenhouse gas emissions. These assessments should serve as bases for improving the formulation of environmentally conditional policies and specific programs, such as initiatives against deforestation and the preservation of the Amazon biome [62].

**Table 1.** Comparison of direct and indirect public policies for the beef industry, the impacts generated, and the costs of these programs for the Brazilian government.

| Public Policy Type | Brazilian Sustainable Agricultural Public Policy | Year Enacted | Program Impacts [Reference] | Program Cost in 2021 USD |
|---|---|---|---|---|
| Direct | Agriculture and Livestock Plan | 2005 | Financing with reduced interest rates [64]. Priority given to technological innovation, storage, irrigation, and low-carbon agriculture [64]. Priority of financing for small- and medium-sized producers [64]. | 48,311,538,461 |
| | ABC Plan or Low Carbon Agriculture | 2010 | Reduce GHG emissions in agriculture [59]. Improve efficiency of natural resource use [59]. Encourage the adoption of Sustainable Production Systems [59]. | 961,000,000 |
| | National Integrated Crop–Livestock–Forest Integration | 2013 | Sustainably improve productivity, product quality, and income from agricultural activities through application of integrated systems [65]. Stimulate research, development, and technological innovation activities [65]. Promote recovery of degraded pasture areas through sustainable production systems [65]. | 13,076,923 |
| | Agriculture Modernization and Natural Resources Conservation Program | 2013 | Soil recovery [52]. Build facilities for the storage of agricultural machinery and implements [52]. Aimed at medium and large rural producers who wish to invest in diverse production [52]. | 68,076,923 |
| In-direct | Brazil Forest Code | 1965 | Declares existing forests as assets of common interest to the entire population and limits the use of rural property by its owners [66]. Maintains and protects permanent preservation areas and on-farm reserves [66]. | - |
| | Action Plan for the Prevention and Control of Deforestation in the Legal Amazon | 2004 | Reduce rate of deforestation in the Amazon [40]. Environmental monitoring and control [40]. Promote sustainable productive activities [40]. | 3,990,384 |
| | Amazon Fund | 2008 | Deforestation reduction with sustainable development in the Amazon [60]. Financing actions for prevention, monitoring, and conservation of Amazon biome [60]. | 337,307,692 |
| | Brazil Forest Code Update | 2012 | Conditions subject to use or management of native vegetation on rural properties [54]. Incentives for technology adoption/good practices reconciling agricultural/forestry productivity with reduced environmental impacts [54]. Recognition of positive impacts in the field in search for sustainable production [54]. | - |

## 5. Future Directions in Agricultural Public Policies for Brazil Beef Production and Sustainability

In the long term, the likely impacts arising from climate change could significantly compromise agricultural activities such as beef cattle production. Some models point to negative scenarios for Brazilian climatic conditions, indicating possible reductions in the availability of water in certain regions and an increase in the availability of water in other regions. In addition to water insecurity, Brazilian agriculture could be impacted by an increase in atmospheric temperatures, which could jeopardize food production and security. These changes may also reduce the profitability of dual cropping systems in Brazil, which are mainly due to shorter rainy seasons, leading to a future shift back to growing only one crop per year instead of two or more [67]. Climate change may also favor nitrous oxide ($N_2O$) emissions if cropping and soil management systems are maintained at their current status [68]. These potential impacts from climate change could result in a negative balance of payments with reductions in products that are destined for export [59].

For the implementation of sustainable production systems, it is first necessary to adopt good agricultural production practices in order to preserve natural resources (e.g., soil, water, biodiversity, and natural forests) that will ensure future production and ecosystem integrity. Combating erosion, recovering degraded soils, and maintaining water sources, natural forests, and biodiversity are priorities that should guide the actions of rural pro-

ducers and frame public policy [21]. Sustainable agricultural systems are more likely to be adopted if they are technically efficient, environmentally suitable, economically viable, and socially accepted [28]. Adequate planning and cost management for the best use of available resources and production factors, with a focus on greater productivity, will become essential for more widespread adoption of economically, socially, and financially sustainable beef cattle production [13].

Over the past decade, new production technologies have been disseminated within Brazil's beef production systems. Technological processes, such as strategic supplementation, semi-confinement, the use of multiple mixtures, genetic crossings, and new forage varieties have enabled Brazil's beef producers to shorten beef production cycles. In addition, technological management methods were incorporated and integrated to reduce production costs and increase economic margins, allowing the beef cattle industry to be one of the more prominent agribusinesses in Brazil [24]. However, the cumulative negative environmental impacts of this increased production of beef have increasingly forced public authorities to question and reconfigure the main notions about food production, which are linked in some ways to forms of development [68].

The role of agriculture in the future could substantially exceed the current traditional systems, requiring joint efforts by the public and private sectors [21]. In order to produce more beef with less environmental impact, Brazil's beef industry must use land more productively. Therefore, it may be necessary to discourage the expansion of speculative, inefficient, and/or riskier agricultural frontiers and to provide services and infrastructure that facilitate investments in areas that have already been deforested [25]. For example, Brazil's final agricultural frontier of Matopiba in northeastern Brazil is drier and more likely not to have enough rainfall for double-cropping (e.g., soybeans followed by corn in the same production year) and not to have adequate groundwater for irrigation [69]. By contrast, agricultural production in Brazil's Amazon and Cerrado (i.e., savannah) biomes can potentially be doubled on existing cleared land without additional deforestation [70].

Assuming the continued implementation of the Brazilian Forest Code and active efforts to re-forest Brazil, beef cattle are projected to increase by 57% on the same amount of pasture, while cropland is expected to increase by 85% by 2050 [71]. Increasing such agricultural production and productivity on land that has already been cleared of forest and native habitat involves sustainable intensification of beef, pasture, and both annual and perennial crops. For Brazil's beef, this can involve reducing greenhouse gas emissions by reducing time to slaughter [72] and by using grain supplementation [18,73]. Pasture productivity in Brazil can be maintained to avoid degradation by re-seeding [18] or through on-farm integration of pasture and commodity crops [74]. Management-intensive rotational grazing (MIRG) was estimated to have about double the carbon removal potential, compared to confined feeding for Brazilian cattle, and MIRG can reduce the pasture area [75] that is required to boost beef productivity. However, MIRG can be more labor-intensive for farmers and farm workers, compared to crop–livestock integration, as cattle have to be frequently rotated between paddocks. Even with public-policy support, farmer adoption may be challenging unless farmers have sufficient resources to commit to the additional labor that is required [74].

Public policies encouraging on-farm integration may be more successful, as producers do not need to coordinate with other farmers. Integration between specialized-crop and livestock farms can be challenging, due to the needs for added coordination and to be close enough to integrate livestock with crops [76,77]. Unlike France, Brazil had limited experience with supra-regional transport of manure and feed over long distances (100 to 500 km) to facilitate crop–livestock integration [78]. Although Brazil has recently increased the semi-confined feeding of beef cattle [79], dairy farms, poultry, and hog farms are typically smaller [80], so manure production does not exceed the capacity of farms to absorb manure nutrients [81]. Additionally, there are no current public policies regulating manure production and application [82]. Despite these challenges, past research suggests that rural credit can be successfully used to incentivize these more complex forms of



integration, such as integrated crop–livestock–forest (ICLF) systems in the Brazilian state of São Paulo [83]. Therefore, the Brazilian government could continue to use the availability of rural credit that is conditional on producers adopting more sustainable agricultural practices. For Brazil's beef producers, the use of rural credit has been associated with reduced deforestation [84].

Such government policies can be easier for agricultural producers to accept if there are clear potential economic benefits, by adopting systems-based approaches such as the ICLF policy or less management-intensive strategies such as sustainable intensification. When designing such recommendations, it is necessary to redirect Brazilian government subsidies to livestock. This guideline is essential in a scenario of budget constraints and current and future climate change [25]. Thus, the attention of national and international organizations and public opinion on illegal deforestation demands intelligence, articulation, and communication in order to guarantee the preservation of natural resources and not to compromise future agricultural exports. Technologies and knowledge are essential factors in promoting sustainable development, mainly by encouraging more sustainable use of resources in the regions where agricultural activities are already concentrated [21]. It is important for public policies to be accessible to rural producers so that Brazil can produce enough food, bioenergy, and commodities for national consumption, while maintaining competitive advantages within the global economy [85].

Although Brazil has abundant water resources, specific commodity-cropping regions, such as Matopiba, face water shortages [69]. Sustainable intensification of crops can involve traditional breeding, especially for maize [86], unlike soybeans. Dry-season irrigation can also increase agricultural output for third crops on the same land base following soybeans and corn [87]. However, Brazilian commodities, such as soybeans and beef cattle, could be sensitive to future drought caused by climate change [88]. Therefore, future agricultural public policies in Brazil can encourage producers to use less water. Due to increasing water scarcity and rising irrigation costs, there has been a growing interest in improving the productivity of water use in agricultural production, with the need to understand the effects of combining water-irrigation management with other agronomic practices for efficient water management and satisfactory yields [89]. Such sustainable intensification of crops can increase environmental sustainability, improve soil conditions, and reduce water pollution. These improvements can potentially benefit the environment, while increasing agricultural productivity [90].

## 6. Conclusions and Implications

The extensive cattle-production system in Brazil is characterized by producers who have been resistant to technological innovations and by the adoption of more intensified management practices. However, in recent years, new technologies have been disseminated in beef production systems in Brazil, such as strategic supplementation, semi-confinement, the use of multiple mixtures of concentrated feeds, genetic crossings, and new forage varieties, which have led to increases in productivity and economic returns. However, the increase in production has generated cumulative environmental impacts, such as deforestation, pasture degradation, and greenhouse gas emissions. Public policies have played an important role in encouraging the adoption of new technologies to mitigate these environmental impacts. However, the impacts caused by beef production in Brazil have forced authorities to develop public policies that allow for increased production but prioritize sustainability, with a focus on the preservation of biomes and natural resources. In order to produce more beef with less environmental impact, it is necessary to encourage livestock farmers to use their land more productively. It is also important to discourage the expansion of speculative and inefficient agricultural frontiers and to provide services and infrastructure that facilitate sustainable agricultural development investments in the regions in Brazil that produce beef cattle. Thus, public policies must continue to be evaluated for their ability to balance agricultural production with resource conservation and environmental preservation.



**Author Contributions:** Conceptualization, L.M., A.K.H. and D.C.d.A.; methodology, A.K.H. and L.M.; validation, L.M. and A.K.H.; formal analysis, L.M., R.A.d.O. and A.K.H.; investigation, L.M.; resources, D.C.d.A., A.K.H. and L.M.; writing—original draft preparation, L.M., A.K.H. and D.C.d.A.; writing—review and editing, all authors; visualization, A.K.H. and R.A.d.O.; supervision, D.C.d.A. and A.K.H.; project administration, D.C.d.A.; funding acquisition, D.C.d.A. and A.K.H. All authors have read and agreed to the published version of the manuscript.

**Funding:** This research was funded by Grupo Osvaldo Sobrinho (Fazenda Boa Vista e Taguá Agropecuária), Serviço Nacional de Aprendizagem Rural de Mato Grosso (SENAR-MT), Programa Global REDD Early Movers de Mato Grosso (REM-MT), and Projeto Rural Sustentável—Cerrado (PRS—Cerrado) e Coordenação de Aperfeiçoamento de Pessoal de Nível Superior (CAPES) for granting a scholarship.

**Institutional Review Board Statement:** Not applicable.

**Informed Consent Statement:** Not applicable.

**Data Availability Statement:** Not applicable.

**Acknowledgments:** We wish to give special thanks to the AgriSciences program at the Federal University of Mato Grosso and Osvaldo Sobrinho for helping with the necessary structure to carry out our research. We also thank the ICAA of the Federal University of Mato Grosso, SENAR-MT, PRS—Cerrado e REM—MT for supporting and encouraging the publication of our study. Finally, we thank three anonymous reviewers whose comments and edits substantially improved the quality of this work.

**Conflicts of Interest:** The authors declare no conflict of interest. Supporting entities had no role in the design of the study; in the collection, analyses, or interpretation of data; in the writing of the manuscript; or in the decision to publish the results.

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
