# Peer review of "Agricultural Support and Public Policies Improving Sustainability in Brazil’s Beef Industry"

_sustainability, doi:10.3390/su15064801_

Round 1

Reviewer 1 Report

The manuscript contains a review about cattle farming in Brazil. The history of cattle breeding and production systems is described in detail. Proposals to increase the sustainability of future cattle farming are outlined.  It is well written, however, this is not a scientific paper, but more a historical/political report. The internationally well reckognized issue of illegal deforestation in Brazil is mentioned only very briefly in line 488 of the manuscript. Readers would probably be interested to learn more specifically how the Brazilian government wants to push through the necessary measures to guarantee a sustainable development in the future. This is currently not outlined in the manuscript. More than 50% of the references are in Brazilian language; this is not appropriate for an article in an international journal.

My judgement "Accept after minor revision" does not include the political aspect of my review.
Detailed comments:

line 58: predominated

line 86: replace "since" by "because"

line 238: either The ABC programs provide or The ABC program provides

line 333: delete "a"

line 530: declare

Author Response

Comments and Suggestions for Authors

The manuscript contains a review about cattle farming in Brazil. The history of cattle breeding and production systems is described in detail. Proposals to increase the sustainability of future cattle farming are outlined.  It is well written, however, this is not a scientific paper, but more a historical/political report. The internationally well recognized issue of illegal deforestation in Brazil is mentioned only very briefly in line 488 of the manuscript.

Readers would probably be interested to learn more specifically how the Brazilian government wants to push through the necessary measures to guarantee a sustainable development in the future. This is currently not outlined in the manuscript.

We have added more writing to the Discussion section in order to clarify how the Brazilian government can successfully insure measures to guarantee more sustainable development in the future. Specifically, we have added two sentences at the end of the third to last paragraph in the Discussion section on continuing to use rural credit conditional on the adoption of more sustainable agricultural practices since for beef producers there is a positive association with tying rural credit to meeting targets for reducing deforestation. We also clarify in the second sentence in the second to last paragraph of the Discussion section that the subsidies discussed are from the Brazilian government.

More than 50% of the references are in Brazilian language; this is not appropriate for an article in an international journal.

We have added 18 journal articles that are not in the Brazilian language. Originally the number of cited references with Portuguese titles was 44 out of 79 (55.7%) with only 35 out of 79 (44.3%) having a title in English. By replacing 10 of 14 Brazilian journal articles with Portuguese titles with 18 journal articles published in international journals, we have increased this to 51 out of 88 (58%) references which are journal articles and a conference paper that have English titles.

My judgement "Accept after minor revision" does not include the political aspect of my review.
Detailed comments:

line 58: predominated

We have corrected this.

line 86: replace "since" by "because"

We have corrected this.

line 238: either The ABC programs provide or The ABC program provides

We have corrected this to the latter.

line 333: delete "a"

We have corrected this.

line 530: declare

We have changed the Data Availability Statement to “Not applicable” since this is a review article.

Reviewer 2 Report

Dear editor,

The manuscript is well-written and has substantial scientific merit. However, there are some minor spelling/grammatical errors that need to be corrected by the authors. In addition, there are some points that the authors should address, and these are mentioned below.

Comments

1.       Add the specific breed type of cattle in the legend of figure 1 for the photograph that you have used. Is it Bos indicus or Bos taurus? If possible, add pasture type?

2.       It would be fine if you provide a brief overview on the different type of beef cattle production system in Brazil especially the different type of extensive cattle production systems in the introduction section.

3.       Line 57 ……………. area 1985 to 2021…………………correct it as    from 1985 to 2021

4.       Line 101 ……………………. Brazil beef systems………………correct as Brazil beef production systems.

5.       Line 189   What is the necessity of capitalization in this phrase ……..”The Agricultural and Livestock Plan”.  Line 192 ……………. Agricultural and Livestock plan……..in this case “p” is small later. Please correct it.

6.       Line 196 ……………. Crop Plan…………apply the former comment.

7.       Change or remove the arrow that you used in Table 1.

8.       Line 278 ……………….. “Moderagro or the Program for the Modernization of Agriculture and Conservation of Natural Resources”…….. no need to write like this if it does not have abbreviation. Similarly check Line 341 ………… Sustainable Development Plan for the Amazon…………………..and also check line 392 …….. Agriculture Modernization and Natural Resources Conservation Program. Line 400 …………. Low Carbon Agricultures….. this is unnecessary capitalization

9.       Line 388 check …………………. Opt…………………? incomplete word?

10.   Inside table 1 you used ……….”greenhouse gas” ………..replace it with GHG since you already abbreviate in the previous pages. The same table you used wrong capitalization for words/terms

11.   In table one the third column ….please correct the term Program impacts………….as Program impacts (References) because you have mention references under this column.

12.   Line 412 ………………title …..Brazil Beef Sustainability…………..insert production next to beef.

13.   Line 496. The first line of conclusion does not give good sense. Please rewrite it.

14.   Did you get permission for figures that you have used in your manuscript?

Best regards

Author Response

The manuscript is well-written and has substantial scientific merit. However, there are some minor spelling/grammatical errors that need to be corrected by the authors. In addition, there are some points that the authors should address, and these are mentioned below.

Comments

  1. Add the specific breed type of cattle in the legend of figure 1 for the photograph that you have used. Is it Bos indicus or Bos taurus? If possible, add pasture type?

We have added Bos indicus Nelore breed and Brachiaria spp. to clarify as requested.

  1. It would be fine if you provide a brief overview on the different type of beef cattle production system in Brazil especially the different type of extensive cattle production systems in the introduction section.

We have added more to the second paragraph in the Introduction section on this.

  1. Line 57 ……………. area 1985 to 2021…………………correct it as    from 1985 to 2021

We have corrected this.

  1. Line 101 ……………………. Brazil beef systems………………correct as Brazil beef production systems.

We have corrected this.

  1. Line 189   What is the necessity of capitalization in this phrase ……..”The Agricultural and Livestock Plan”.  Line 192 ……………. Agricultural and Livestock plan……..in this case “p” is small later. Please correct it.

We have chosen to keep the P capitalized. We have corrected this to make consistent with elsewhere in the manuscript by capitalizing the P and using “Agriculture” instead of “Agricultural.”

  1. Line 196 ……………. Crop Plan…………apply the former comment.

      We have chosen to also keep the P capitalized for Crop Plan.

  1. Change or remove the arrow that you used in Table 1.

We have removed all arrows in Table 1.

  1. Line 278 ……………….. “Moderagro or the Program for the Modernization of Agriculture and Conservation of Natural Resources”…….. no need to write like this if it does not have abbreviation.

      We have changed this from italics to regular font to be consistent with other sub-section titles. We feel it is important to include the Portuguese wording for the programs.

Similarly check Line 341 ………… Sustainable Development Plan for the Amazon…………………..

We not capitalized this.

and also check line 392 …….. Agriculture Modernization and Natural Resources Conservation Program.

      This is capitalized in Table 1 so we kept this as is.

Line 400 …………. Low Carbon Agricultures….. this is unnecessary capitalization

We have changed this by getting rid of the capitalization.

  1. Line 388 check …………………. Opt…………………? incomplete word?

      We have deleted the word “opt” and written this more clearly.

  1. Inside table 1 you used ……….”greenhouse gas” ………..replace it with GHG since you already abbreviate in the previous pages. The same table you used wrong capitalization for words/terms

      We have used GHG.

  1. In table one the third column ….please correct the term Program impacts………….as Program impacts (References) because you have mention references under this column.

      We have used [Reference].

  1. Line 412 ………………title …..Brazil Beef Sustainability…………..insert production next to beef.

      We have written as “…Brazil Beef Production Sustainability.”

  1. Line 496. The first line of conclusion does not give good sense. Please rewrite it.

      We have re-written this sentence as “The extensive cattle production system in Brazil is characterized by producers who have been resistant to technological innovations and the adoption of more intensified management practices.”

  1. Did you get permission for figures that you have used in your manuscript?

The data we used from Brazil government agencies is publicly available for use by anyone and we have secured confirmations from these agencies that we are allowed to use these data.

Best regards

Thank-you very much for your constructive comments and edits which has improved the quality of this manuscript!

Reviewer 3 Report

This review provides an overview of beef production in Brazil, particularly on the historical factors that have encouraged an extensive, low-intensity style of production. How national agricultural public policies have improved sustainability in Brazil’s beef industry was also discussed. It is interesting and could be accepted after a major revision.

Line 23: please add a reference to certify this information.

Line 39: Please add some description for the cattle breeds that are reared in Brazil and their economical values.

Line 52: Could you clarify whether the current annual production is enough or not for the Brazl population?

Line 58-67: I think this part has no major value. Try to shorten it, please.

Line 70: new technologies were referred to what? Please clarify.

Line 85: The authors should have the rights or permission to use these figures?

Line 90-93: please rephrase this sentence to be clearer.

Line 94: please try to improve your rationale. What are the outcomes after addressing these points in your literature?

Line 103-106: Please explain the reason that Brazilian beef cattle producers are historically characterized by resistance to technological innovations and more primitive management.

Line 123-129 should be combined with the previous paragraph. Also, line 130-138

Line 139-172 are superfluous and not organized well. Please combine these lines in a single paragraph and avoid elongation.

Line 175-176: during the last decade or during which time? Please clarify here.

Line 174-288: I also suggest summarizing it in a table or a chart. Also lines 289-363

Line 364: it should be not included in the contents of the subtitle (3. Recent Public Policies for More Sustainable Livestock in Brazil). Or I recommend editing this subtitle. The authors should highlight after line 173 that there were two public policies; direct and indirect ones, for getting More Sustainable Livestock in Brazil.

Line 403. This table is very good. Why the authors did not organize it chronologically?

Line 413-414: please explain more about how climate change could significantly compromise agricultural activities in Brazil.

Line 485 such recommendations are referred to what? Please combine the paragraphs and try to organize your information in the whole manuscript.

Line 497-499: please rephrase this sentence.

Line 496: It should be the conclusion and implications.

Author Response

Comments and Suggestions for Authors

This review provides an overview of beef production in Brazil, particularly on the historical factors that have encouraged an extensive, low-intensity style of production. How national agricultural public policies have improved sustainability in Brazil’s beef industry was also discussed. It is interesting and could be accepted after a major revision.

Line 23: please add a reference to certify this information.

MDPI does not use [#] citations in the Abstract. This sentence in the abstract is similar to the following sentence toward the end of the Introduction section which is cited: “Such strategies will also result in a significant net reduction in greenhouse gas emissions [28].”

Line 39: Please add some description for the cattle breeds that are reared in Brazil and their economical values.

We have added two sentences in the first paragraph of the Introduction section describing better tropical Bos indicus Nelore and temperate Bos taurus beef cattle in Brazil with citations.

Line 52: Could you clarify whether the current annual production is enough or not for the Brazil population?

We have clarified this in a sentence added to the middle of the third paragraph of the Introduction.

Line 58-67: I think this part has no major value. Try to shorten it, please.

We have shortened and re-written this paragraph to:

“The first cattle arrived in Brazil in 1533, during the establishment of the first Portuguese colony on the island of São Vicente in the state of São Paulo [6]. In the middle of the 16th century, the Portuguese royal court encouraged the export of cattle to the Bahian Recôncavo region in northeastern Brazil. Gradually, with the growth of the economy in coastal areas, cattle raising expanded into the country’s interior [7]. Since these commercial beginnings, Brazil’s beef production has relied on an extensive pasture-based system using plants adapted to local climate and soil conditions with limited use of inputs [8].”

Line 70: new technologies were referred to what? Please clarify.

We have replaced “new technologies” with “practices” to describe this better.

Line 85: The authors should have the rights or permission to use these figures?

The data we used from Brazil government agencies is publically available for use by anyone and we have secured confirmations from these agencies that we are allowed to use these data.

Line 90-93: please rephrase this sentence to be clearer.

We have shortened this sentence to “Amazon rainforest deforestation preceding establishment of pastures has focused international attention on reducing deforestation [4].”

Line 94: please try to improve your rationale. What are the outcomes after addressing these points in your literature?

In order to improve our rationale, we have added more to an existing sentence and we have added a concluding sentence to the last paragraph of the Introduction section.

Line 103-106: Please explain the reason that Brazilian beef cattle producers are historically characterized by resistance to technological innovations and more primitive management.

We have explained this as a preference for management systems that take less time at the end of the second to last paragraph in the Introduction section.

Line 123-129 should be combined with the previous paragraph. Also, line 130-138

We have merged the 2nd and 3rd paragraphs of section 2 as requested.

Line 139-172 are superfluous and not organized well. Please combine these lines in a single paragraph and avoid elongation.

We have reduced the last 4 paragraphs in this second section to 2 paragraphs by eliminating superfluous writing and focusing the organization better.

Line 175-176: during the last decade or during which time? Please clarify here.

We have clarified this by adding a sentence in the beginning of the first paragraph of section 3.1.

Line 174-288: I also suggest summarizing it in a table or a chart. Also lines 289-363

We have summarized the timeline in Figure 6 and in table form in Table 1.

Line 364: it should be not included in the contents of the subtitle (3. Recent Public Policies for More Sustainable Livestock in Brazil). Or I recommend editing this subtitle. The authors should highlight after line 173 that there were two public policies; direct and indirect ones, for getting More Sustainable Livestock in Brazil.

We have made this into a new section: “4. Comparison of Agricultural Public Policies” and updated numbers of sections 5 and 6 that follow accordingly.

We have added an introductory paragraph to Section 3 as recommended: “We discuss two types of recent sustainable agricultural development policies in Brazil. In section 3.1., we cover agricultural policies that have had a direct impact on improving sustainable intensification (SI) in Brazil’s beef production industry. In section 3.2., we highlight environmental public policies that have had a more indirect impact on the SI of Brazil’s beef cattle herd. These environmental policies have typically preceded the more direct policies in section 3.1. and have involved reducing Amazon deforestation and land use conversion of other natural habitat in Brazil for cattle pasture. ”

Line 403. This table is very good. Why the authors did not organize it chronologically?

We have chosen to keep the direct and indirect parts of the table separate and within these two categories we have chronologically ordered as you suggested.

Line 413-414: please explain more about how climate change could significantly compromise agricultural activities in Brazil.

We have added two sentences explaining more how climate change could significantly compromise Brazil agriculture in the middle of the first paragraph of the Discussion section.

Line 485 such recommendations are referred to what? Please combine the paragraphs and try to organize your information in the whole manuscript.

We have reorganized this part of the Discussion section so that we end (last paragraph in revised Discussion section) on discussion of the need for future government agricultural polices to address water conservation since none of the current policies that we discuss direct address sustainable water use.

Line 497-499: please rephrase this sentence.

We have re-written the first sentence of the first paragraph in the Conclusion section to read “The extensive cattle production system in Brazil is characterized by producers who have been resistant to technological innovations and the adoption of more intensified management practices.”

Line 496: It should be the conclusion and implications.

We have renamed the title as suggested to “6. Conclusions and Implications” since this also covers implications.

Round 2

Reviewer 3 Report

Thank you very much for addressing most comments in the first round of reviewing. Your manuscript should be accepted after addressing these minor comments:

1-Line 76: The incorporation of practices should be clarified. Also in the whole manuscript.

2-Please add more description for the tables and figures.

Author Response

Comments and Suggestions for Authors

Thank you very much for addressing most comments in the first round of reviewing. Your manuscript should be accepted after addressing these minor comments:

1-Line 76: The incorporation of practices should be clarified. Also in the whole manuscript.

We clarified this in line 87 and elsewhere in the text where this was necessary. We also cite examples of these practices in the summary and conclusion section.

2-Please add more description for the tables and figures.

We have added more details to the descriptions for Figures 1, 2, 4 as well as for Table 1.

Submission Date

30 January 2023

Date of this review

27 Feb 2023 20:50:00